# Evaluation of models for prognosing mortality in critical care patients with COVID-19: First- and second-wave data from a German university hospital

**Martin Kieninger**[1]*, **Sarah Dietl**[1], **Annemarie Sinning**[1], **Michael Gruber**[1], **Wolfram Gronwald**[2], **Florian Zeman**[3], **Dirk Lunz**[1], **Thomas Dienemann**[4], **Stephan Schmid**[5], **Bernhard Graf**[1], **Matthias Lubnow**[6], **Thomas Müller**[6], **Thomas Holzmann**[7], **Bernd Salzberger**[7], **Bärbel Kieninger**[7]

1 Department of Anesthesiology, University Medical Center Regensburg, Regensburg, Germany, 2 Institute of Functional Genomics, University of Regensburg, Regensburg, Germany, 3 Center for Clinical Studies, University Medical Center Regensburg, Regensburg, Germany, 4 Department of Surgery, University Medical Center Regensburg, Regensburg, Germany, 5 Department of Internal Medicine I, University Medical Center Regensburg, Regensburg, Germany, 6 Department of Internal Medicine II, University Medical Center Regensburg, Regensburg, Germany, 7 Department of Infection Prevention and Infectious Diseases, University Medical Center Regensburg, Regensburg, Germany

* martin.kieninger@ukr.de

**Data Availability Statement:** All relevant data are within the article and its Supporting Information files.

## Abstract

### Background

In a previous study, we had investigated the intensive care course of patients with coronavirus disease 2019 (COVID-19) in the first wave in Germany by calculating models for prognosticating in-hospital death with univariable and multivariable regression analysis. This study analyzed if these models were also applicable to patients with COVID-19 in the second wave.

### Methods

This retrospective cohort study included 98 critical care patients with COVID-19, who had been treated at the University Medical Center Regensburg, Germany, between October 2020 and February 2021. Data collected for each patient included vital signs, dosage of catecholamines, analgosedation, anticoagulation, and antithrombotic medication, diagnostic blood tests, treatment with extracorporeal membrane oxygenation (ECMO), intensive care scores, ventilator therapy, and pulmonary gas exchange. Using these data, expected mortality was calculated by means of the originally developed mathematical models, thereby testing the models for their applicability to patients in the second wave.

### Results

Mortality in the second-wave cohort did not significantly differ from that in the first-wave cohort (41.8% vs. 32.2%, p = 0.151). As in our previous study, individual parameters such as pH of blood or mean arterial pressure (MAP) differed significantly between survivors and

**Funding:** The authors received no specific funding for this work.

**Competing interests:** The authors have declared that no competing interests exist.

non-survivors. In contrast to our previous study, however, survivors and non-survivors in this study showed significant or even highly significant differences in pulmonary gas exchange and ventilator therapy (e.g. mean and minimum values for oxygen saturation and partial pressure of oxygen, mean values for the fraction of inspired oxygen, positive expiratory pressure, tidal volume, and oxygenation ratio). ECMO therapy was more frequently administered than in the first-wave cohort. Calculations of expected mortality by means of the originally developed univariable and multivariable models showed that the use of simple cut-off values for pH, MAP, troponin, or combinations of these parameters resulted in correctly estimated outcome in approximately 75% of patients without ECMO therapy.

## Introduction

Since its first identification in 2019, infection with coronavirus disease 2019 (COVID-19) has proceeded in waves, not only in Germany but all over the world. In Germany, the first wave reached its peak in April 2020, and the second wave peaked in December 2020 [1]. These waves were primarily triggered by the spread of different virus variants [2–4].

In a previous study, we had examined in detail the first two weeks of intensive care therapy in patients with COVID-19 treated in one of the intensive care units (ICU) at the University Medical Center Regensburg during the first wave. Factors predicting hospital mortality were identified by means of univariable and multivariable analysis. Blood pH, mean arterial pressure (MAP), base excess (BE), troponin, and procalcitonin were found to be of particular relevance in this context [5].

Many research projects have been initiated worldwide to optimize the treatment of critical care patients with COVID-19. For example, treatment with dexamethasone has become standard therapy since the publication that ICU patients benefit from this medication [6]. Accordingly, since the publication of this finding, ICU patients have received this therapy regularly, whereas previously, based on the state of knowledge at that time, the administration of corticosteroids was only recommended in cases of refractory shock [7].

Due to intervening changes in the therapeutic approach, the courses of disease of ICU patients in the second wave were likely to differ from that of ICU patients in the first wave. Consequently, differing ICU courses may also result in different prognostic factors for unfavorable outcome.

## Material and methods

### Ethics approval and consent to participate

The study was approved by and conducted according to the guidelines of the Ethics Committee of the University of Regensburg (approval number 20-1790-104, 'S1 Appendix'). In accordance with European law, consent to participate was not required because of the retrospective study design and the use of anonymized patient data. All data were anonymized prior to analysis.

### Aim of the study

Analogous to the procedure of investigating patients of the first wave, this study evaluated the intensive care course of all patients treated for COVID-19 at one of our intensive care units during the second wave. First, we identified any differences in the collected parameters

between survivors and non-survivors in the second-wave cohort and compared the results with those of the previous study. Of particular interest was whether the models for prognosing unfavorable outcome calculated for the patients in the first-wave cohort also applied to the patients in the second-wave cohort.

## Patients and settings

This retrospective study included newly acquired data from 98 critically ill adult patients with COVID-19 (76 men, 22 women), who had been treated at one of the ICUs at the University Medical Center Regensburg between October 2020 and February 2021 (second-wave) and comparative data from 59 patients of the first wave, as already published in [5]. The cohorts consisted of patients either directly admitted to the University Medical Center Regensburg or transferred from a non-tertiary hospital in the surrounding area for higher care therapy. There were no standardized criteria for the admission of COVID-19 patients to an ICU at the University Medical Center Regensburg. In the present study, 57 patients were discharged from the ICU and 41 had died, yielding a mortality rate of 41.8% compared to the mortality rate of 32.2% found in the previous study. Mortalities did not differ significantly between the two cohorts (p = 0.151). Fig 1 shows the Kaplan-Meier estimator for the second-wave cohort.

## Data collection

We examined the first two weeks of ICU treatment or the time until a patient had died or was discharged from the ICU. Data collection was analogous to the procedure used in the previous

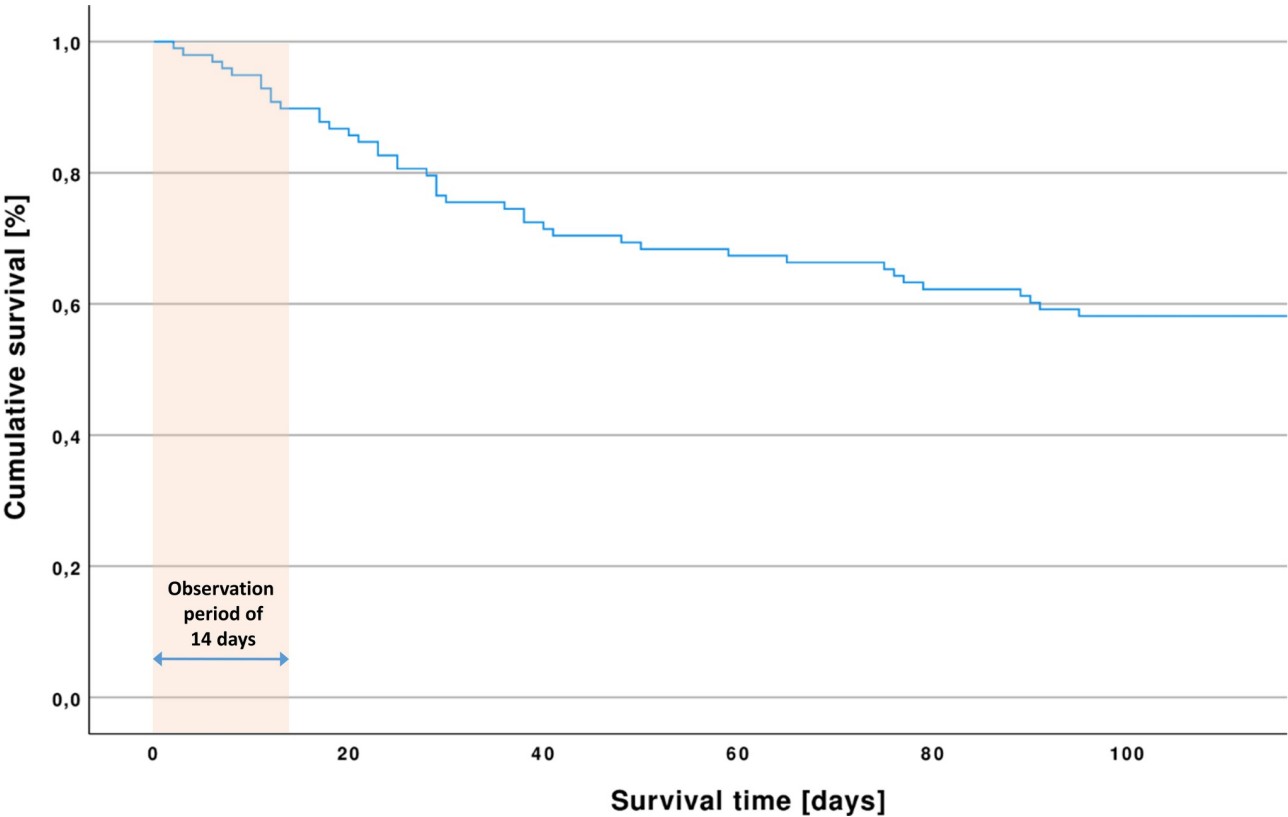

**Fig 1. Kaplan-Meier estimator for all patients of the second-wave cohort.** Out of 98 patients included in the study, 41 did not survive. 10 patients had died within the observation period of the first 14 days of intensive care treatment, and 57 patients were discharged from the intensive care unit.

study. The complete list of all parameters examined is provided in 'S1 Table.' Complete data sets consisted of 883 values per patient. Data were extracted from the data management systems of the ICUs (MetaVisionSuite®, version V6.9.0.23, iMDsoft®, Tel Aviv, Israel; SAP® Enterprise resource planning, version 6.0 EHP7 SP21, SAP SE, Walldorf, German; SWISSLAB® Laborinformationssysteme, version 2.18.3.00, NEXUS SWISSLAB GmbH, Berlin, Germany). The patients were grouped according to their outcome (died: death in intensive care, survived: transferred to a rehabilitation or general care unit); outcome was determined on the day of discharge from the ICU.

## Statistical analysis

Statistical analysis was conducted using IBM SPSS Statistics™ 28 (IBM, Armonk, USA). Statistical tests were two-sided, and the level of significance was set to $p < 0.050$ (termed 'significant'). Categorical parameters are presented as absolute and relative frequencies and shown as bar charts. Data of survivors and non-survivors were compared with the Chi-square test of independence. Continuous data are shown as median and minimum/maximum, and differences between survivors and non-survivors were assessed with the Mann-Whitney-U test. Continuous parameters with values for each day of examination were analyzed in two steps. First, the course within the first 14 days of ICU treatment was investigated by calculating the median and interquartile range for survivors and non-survivors for each day. Both groups were compared using the two-sided Man-Whitney-U test. Results are visualized by means of box plots. Second, mean values and, if reasonable from a clinical point of view, minimum and maximum values were calculated for each patient over the course of time, which were then compared between survivors and non-survivors using the Mann-Whitney-U test ('S2 Table'). To account and correct for multiple comparisons, Bonferroni correction was applied, and a newly significance level $p^* = 5.43^* 10^{-4}$ (termed 'highly significant') was determined. In S2 Table, data from the first wave were also included, and differences between survivors and non-survivors were assessed by categorization across the defined significance levels. Finally, the cut-off values for the parameters previously identified as highly significant by means of univariable regression analysis in the first-wave cohort were used to form two groups with respect to each of these parameters and to calculate mortality rates of the second-wave cohort. This procedure was analogously performed for multivariable regression analysis ('S2 Appendix').

## Results

### Baseline and demographic data of the second-wave cohort

In the present study, survivors had a median age of 56 years (minimum 22 years, maximum 81 years). Non-survivors had a median age of 62 years (minimum 44 years, maximum 80 years) and were thus significantly older (p = 0.002). Age did not differ between the second-wave cohort and the first-wave cohort (median age of second-wave cohort 58 years, median age of first-wave cohort 60 years, p = 0.846). The second-wave cohort consisted of 76 men and 22 women. 16 women and 41 men survived, and 6 women and 35 men had died (p = 0.144). The body mass index did not differ significantly between survivors and non-survivors (survivors: median 27.8 kg/m$^2$, minimum 18.4 kg/m$^2$, maximum 52.1 kg/m$^2$; non-survivors: median 29.3 kg/m$^2$, minimum 23.1 kg/m$^2$, maximum 66.7 kg/m$^2$; p = 0.386).

Survivors received ICU treatment for a median of 23 days (minimum 2 days, maximum 82 days), non-survivors for a median of 29 days (minimum 2 days, maximum 95 days). Survivors in the second-wave cohort received significantly shorter ICU treatment (p = 0.031) and non-survivors significantly longer ICU treatment (p = 0.016) than patients of the first-wave cohort (median time of ICU treatment for survivors 31 days, for non-survivors 17 days). 73% of the

survivors as well as 73% of the non-survivors in the second-wave cohort had been initially treated at an external ICU before being transferred to an ICU at the University Medical Center Regensburg. Treatment duration before the transfer did not differ between survivors and non-survivors (median 3 days, maximum 47 days vs. median 8 days, maximum 36 days; p = 0.116). In comparison to the patients of the first-wave cohort, of whom 63% had initially been treated at an external ICU (survivors median 3 days, non-survivors median 4 days; p = 0.036), patients of the second-wave cohort had been treated significantly longer at an external ICU prior to their transfer to the University Medical Center Regensburg.

Pre-existing comorbidities were categorized as cardiovascular, pneumological, autoimmune, oncological, neurological, infectious, nephrological, and degenerative diseases, as obesity or diabetes mellitus, and as being caused by other metabolic, allergenic, and noxious substances. On average, patients of the second-wave cohort had pre-existing diseases from three categories, patients of the first-wave cohort only from two categories (p = 0.020). Most patients with comorbidity had cardiovascular diseases (second-wave cohort 68.4%, first-wave cohort 52.5%). Comparing survivors to non-survivors in each category, no significant differences were detected between patients of the second-wave and the first-wave cohort.

## Course of parameters relevant for ICU treatment during the observation period

Daily records of metric and categorical parameters are graphically presented in 'S1–S5 Figs' to show their course over the 14-day observation period. 'S2 Table' shows a comparison of mean values for metric parameters, and, if reasonable, the maximum and minimum values of survivors and non-survivors of the first-wave and second-wave cohort.

## Vital signs and intensive care scores ('S1 Fig', 'S2 Table')

Body temperature was recorded daily as a categorical variable with two possible values 'fever' (daily temperature peaks of $\geq 38.0°C$) and 'no fever' (temperature $< 38.0°C$). The percentage of non-survivors with fever was significantly higher than that of non-survivors on 6 out of the 14 days of the observation period.

Within the first week of the observation period, daily heart rate (HR) was higher in non-survivors than in survivors. Mean HR values over the entire observation period differed significantly between survivors (74.8 bpm) and non-survivors (81.0 bpm) (p = $1.30^*10^{-2}$).

Survivors showed significantly higher oxygen saturation ($SpO_2$) on each of the 14 days as well as significantly higher mean and minimum $SpO_2$ values over the entire observation period (mean: 95.0% for survivors vs. 93.5% for non-survivors, p = $2.91^*10^{-5}$, minimum: 92.8% for survivors vs. 90.1% for non-survivors, p = $1.06^*10^{-5}$). Survivors and non-survivors in the first-wave cohort had neither differed in mean nor in minimum oxygen saturation.

Mean arterial pressure (MAP) was significantly higher in survivors than in non-survivors on 10 out of the 14 days of the observation period. Over the entire observation period, mean MAP was significantly different between survivors and non-survivors in the second-wave cohort and highly significantly different in the first-wave cohort (second-wave cohort: 82.3 mmHg vs. 77.1 mmHg, p = $7.43^*10^{-4}$; first-wave cohort: 81.7 mmHg vs. 74.1 mmHg, p = $4.88^*10^{-5}$). For minimum MAP, the difference was highly significant in the second-wave cohort and significant in the first-wave cohort (second-wave cohort: 72.5 mmHg vs. 70.0 mmHg, p = $2.62^*10^{-5}$; first-wave cohort: 92.7 mmHg vs. 84.2 mmHg, p = $3.15^*10^{-3}$).

The cumulative mean value for the Therapeutic Intervention Scoring System (TISS) was significantly higher for non-survivors (TISS: 12.0 for survivors vs. 14.7 for non-survivors, p = $1.64^*10^{-3}$). The difference in the Simplified Acute Physiology Score (SAPS) between non-

survivors and survivors was highly significant (SAPS: 29.5 for non-survivors vs. 21.2 for survivors, p = $9.80^*10^{-7}$). These results were in line with the findings for patients of the first-wave cohort. However, a shift towards lower TISS and SAPS scores was noted in the second-wave cohort in comparison to the first-wave cohort.

### Dosage of catecholamines, analgosedation, anticoagulation, and antithrombotic medication ('S2 Fig', 'S2 Table')

The mean hourly dose of norepinephrine per day was significantly higher for non-survivors than for survivors on each day of the observation period. Over the entire observation period, differences in the mean and maximum values of the hourly dose of norepinephrine per day between survivors and non-survivors were highly significant in the second-wave cohort (mean: 0.10 mg/h for survivors, 0.24 mg/h for non-survivors, p = $4.58^*10^{-5}$, maximum: 0.35 mg/h for survivors, 0.73 mg/h, p = $4.31^*10^{-4}$) and significant in the first-wave cohort (mean: 0.27 mg/h for survivors, 0.61 mg/h for non-survivors, p = $2.28^*10^{-3}$, maximum: 0.73 mg/h for survivors, 1.50 mg/h, p = $8.81^*10^{-4}$).

Non-survivors in the second-wave cohort received a significantly higher mean daily dosage of sufentanil than survivors on each day of the observation period; no or only marginal differences were found for propofol, midazolam, and ketamine.

Most patients were treated with doses of unfractionated or low molecular weight heparin that were higher than the prophylactic doses; there were no significant differences between survivors and non-survivors, but non-survivors had more often received acetylsalicylic acid.

### Diagnostic blood tests ('S3 Fig', 'S2 Table')

Non-survivors in the second-wave cohort had significantly lower daily pH of blood values on 9 days of the observation period, highly significantly lower mean values over the entire observation period, and significantly lower minimum values (mean: 7.438 for survivors, 7.405 for non-survivors, p = $3.63^*10^{-4}$, minimum: 7.368 for survivors, 7.321 for non-survivors, p = $5.68^*10^{-4}$). Patients of the first-wave cohort had shown highly significant differences in both mean and minimum pH of blood values (mean: 7.422 for survivors, 7.344 for non-survivors, p = $5.47^*10^{-8}$, minimum: 7.350 for survivors, 7.250 for non-survivors, p = $1.30^*10^{-7}$). Regarding BE, blood bicarbonate, blood lactate, and blood chloride values, no relevant differences were found between survivors and non-survivors in the second-wave cohort. In contrast, survivors and non-survivors in the first-wave cohort had shown significant differences in mean values of blood bicarbonate and highly significant differences in mean BE values.

Survivors in the second-wave cohort had higher daily mean values for arterial partial pressure of oxygen (paO$_2$) on 6 days and lower daily mean values for arterial partial pressure of carbon dioxide (paCO$_2$) on 13 out of the 14 days of the observation period. Survivors and non-survivors differed significantly in the cumulative mean value for paO$_2$ and highly significantly in the cumulative mean value for paCO$_2$ (paO$_2$: 81.9 mmHg vs. 77.6 mmHg, p = $5.11^*10^{-3}$, paCO$_2$: 41.5 mmHg vs. 47.4 mmHg, p = $1.03^*10^{-4}$). The difference was significant for mean paO$_2$ (68.3 mmHg vs. 65.3 mmHg, p = $1.79^*10^{-2}$) and even highly significant for mean maximum of paCO$_2$ (49.0 mmHg vs. 59.5 mmHg, p = $3.11^*10^{-5}$). In the first-wave cohort, survivors and non-survivors had significantly differed in paCO$_2$ (mean paCO$_2$ 43.9 mmHg vs. 49.2 mmHg, p = $9.14^*10^{-3}$; mean minimum paCO$_2$ 35.1 mmHg vs. 39.2 mmHg, p = $3.51^*10^{-2}$; mean maximum paCO$_2$ 52.7 mmHg vs. 62.9 mmHg, p = $5.54^*10^{-3}$).

There was a highly significant difference in cumulative mean and maximum values of troponin T between survivors and non-survivors (mean: 48.0 ng/L vs. 124.5 ng/l, p = $3.31^*10^{-6}$, maximum: 77.8 ng/l vs. 278.2 ng/l, p = $5.40^*10^{-5}$). In the first-wave cohort the differences ware

highly significant for overall mean troponin T and significant for the maximum values (mean: 35.4 ng/L vs. 246.9 ng/l, p = $2.78^{*}10^{-4}$, maximum: 66.4 ng/l vs. 369.6 ng/l, p = $1.41^{*}10^{-3}$).

Regarding the cumulative mean values for the remaining parameters of the diagnostic blood tests, survivors and non-survivors in this study differed highly significantly in values for urea (70.2 mg/dL for survivors vs. 101.3 mg/dL for non-survivors, p = $4.99^{*}10^{-4}$), international normalized ratio (INR, 1.10 for survivors vs. 1.31 for non-survivors, p = $5.44^{*}10^{-5}$), C-reactive protein (CRP, 80.1 mg/L for survivors vs. 122.7 mg/L for non-survivors, p = $3.27^{*}10^{-4}$), and interleukin 6 (IL-6, 117.8 pg/mL for survivors vs. 367.2 pg/mL for non-survivors, p = $9.93^{*}10^{-5}$). Survivors and non-survivors differed significantly in cumulative mean values for glomerular filtration rate (eGFR, 84.9 mL/min/KOF for survivors vs. 66.0 mL/min/KOF for non-survivors, p = $8.02^{*}10^{-3}$), creatinine (1.11 mg/dL for survivors vs. 1.50 mg/dL for non-survivors, p = $4.88^{*}10^{-2}$), aspartate transaminase (AST, 88.1 U/L for survivors vs. 287.1 U/L for non-survivors, p = $1.66^{*}10^{-3}$), alanine transaminase (ALT, 78.5 U/L for survivors vs. 190.5 U/L for non-survivors, p = $6.21^{*}10^{-3}$), procalcitonin (PCT, 1.05 ng/mL for survivors vs. 2.75 ng/ml for non-survivors, p = $8.04^{*}10^{-4}$), ferritin (1848 ng/mL for survivors vs. 3362 for non-survivors, p = $3.82^{*}10^{-2}$), D-dimers (8.0 mg/L for survivors vs. 12.4 mg/L for non-survivors, p = $3.33^{*}10^{-2}$), and platelet count (254/nL for survivors vs. 186/nL for non-survivors, p = $2.55^{*}10^{-3}$). In contrast, survivors and non-survivors in the first-wave cohort had not significantly differed in cumulative mean values for INR, ALT, ferritin, D-dimers, and platelets, but non-survivors had shown a significantly higher white blood cell count than survivors.

## Ventilator therapy ('S4 Fig', 'S2 Table')

Non-survivors in the second-wave cohort needed significantly higher fractions of inspired oxygen (FiO$_2$) on 12 days and positive end expiratory pressure (PEEP) on 11 out of 14 days of the observation period than survivors. In contrast, tidal volume was lower in non-survivors on 13 days and the paO$_2$/FiO2 ratio (P/F ratio) on 11 out of 14 days. Correspondingly, the cumulative mean values for these parameters differed significantly or even highly significantly between survivors and non-survivors (FiO$_2$: 53.7% for survivors vs. 63.4% for non-survivors, p = $4.11^{*}10^{-4}$, PEEP: 10.3 mmHg for survivors vs. 12.5 mmHg for non-survivors, p = $1.71^{*}10^{-3}$, tidal volume: 456 mL for survivors vs. 340 mL for non-survivors, p = $1.20^{*}10^{-3}$, P/F ratio: 174 for survivors vs. 138 for non-survivors, p = $4.05^{*}10^{-4}$). These results contrasted with the findings for the patients of the first-wave cohort, in which survivors and non-survivors had not significantly differed in ventilator parameters.

With regard to daily kinetic therapy with proning, no statistically detectable difference was found between survivors and non-survivors in this study.

## Extracorporeal membrane oxygenation (ECMO) and renal replacement therapy (RRT) ('S5 Fig')

In this study, ECMO therapy was administered to 50% of the patients (49 out of 98) and to more non-survivors than survivors (27 out of 41 patients, 66% vs. 22 out of 57 patients, 39%, p = 0.014). For further examination, patients were grouped into the following categories: start of ECMO on day 1 to 7, start of ECMO on day 8 to 14, duration of ECMO <5 days, and duration of ECMO ≥5 days. Almost all patients received ECMO therapy within the first week of ICU treatment; only one non-survivor received ECMO therapy in the second week. ECMO therapy was administered for at least 5 days to 20 out of 22 patients who survived and to 25 out of 27 non-survivors. Patients in the first-wave cohort had been less frequently treated with ECMO therapy (49 out of 98 patients, 50% of patients in the second-wave cohort vs. 17 of 59 patients, 29% of patients in the first-wave cohort, p = 0.012).

Significantly more non-survivors than survivors received RRT (15 out of 41 patients, 37% vs. 8 out of 57 patients, 14%, p = 0.015). This result was in line with the findings for the first-wave cohort. RRT was less frequently required in the second-wave cohort (23 out of 98 patients, 23% of patients in the second-wave cohort vs. 25 of 59 patients, 42% of patients in the first-wave cohort, p = 0.020).

## Evaluation of the models calculated for the first-wave cohort

Table 1 shows the results of testing the univariable models for prognosing a lethal course, determined by means of patient data of the first-wave cohort with the values of the patients of the second-wave cohort. These calculations were additionally performed with the exclusion of patients with ECMO therapy. In first-wave patients included in the previous study, relative mortality was 2.64 for MAPmean, 7.89 for pHmean, 3.57 for pHmax, 5.03 for pHmin, 2.57 for BEmean, 3.25 for BEmax, and 2.88 for troponin Tmean.

As in the first-wave cohort, pH of blood was the parameter with the highest prognostic power. Blood pH, MAP, and troponin showed a clearer separation between the groups when only considering patients without ECMO therapy. The outcome of approximately 75% of patients (MAPmean 69.4%, pHmean 75.5%, pHmax 75.5%, pHmin 77.6%, and Troponin Tmean 71.7%) could have been correctly prognosed in these patients. BE, on the other hand, was no longer suitable as a prognosing factor for lethal outcome, even after excluding patients with ECMO therapy.

Using the multivariable model generated in the previous study, a cut-off value of 7.93 was calculated from the parameters MAPmean and pHmin by setting the probability of death to

**Table 1. Relative mortality in the second-wave cohort (univariable analysis).** Using the cut-off values for a survival probability of 50%, which had been determined for the patients of the first-wave cohort by univariable regression analysis, the second-wave cohort was divided into two groups: patients with values above and patients with values below the cut-off. Mortalities within the groups and, from these results, the relative mortality was calculated. MAPmean, mean MAP during the 14-day observation period for each patient; pHmean/pHmax/pHmin, mean, maximum, and minimum pH of blood during the 14-day observation period for each patient; BEmean/BEmax, mean and maximum BE during the 14-day observation period for each patient; Troponin Tmean, mean troponin T during the 14-day observation period for each patient.

| | | Cut-off | Absolute mortality of patients with values below (MAP, pH, BE) or above (Troponin) cut-off | Absolute mortality of patients with values above (MAP, pH, BE) or below (Troponin) cut-off | Relative mortality |
|---|---|---|---|---|---|
| **MAPmean** | All patients | 75 mmHg | 12/22 patients (54.5%) | 29/76 patients (38.2%) | 1.43 |
| | Only patients without ECMO | | 4/9 patients (44.4%) | 10/40 patients (25.0%) | 1.78 |
| **pHmean** | All patients | 7.38 | 12/17 patients (70.1%) | 29/81 patients (35.8%) | 1.97 |
| | Only patients without ECMO | | 8/14 patients (57.1%) | 6/35 patients (17.1%) | 3.33 |
| **pHmax** | All patients | 7.44 | 7/9 patients (77.8%) | 34/89 patients (38.2%) | 2.04 |
| | Only patients without ECMO | | 5/8 patients (62.5%) | 9/41 patients (22.0%) | 2.85 |
| **pHmin** | All patients | 7.28 | 10/13 patients (76.9%) | 31/85 patients (36.5%) | 2.11 |
| | Only patients without ECMO | | 6/9 patients (66.7%) | 8/40 patients (20.0%) | 3.33 |
| **BEmean** | All patients | -0.59 mmol/L | 4/10 patients (40.0%) | 37/88 patients (42.0%) | 0.95 |
| | Only patients without ECMO | | 2/7 patients (28.6%) | 12/42 patients (28.6%) | 1.00 |
| **BEmax** | All patients | 2.68 mmol/L | 5/12 patients (41.7%) | 36/86 patients (41.9%) | 1.00 |
| | Only patients without ECMO | | 2/8 patients (25.0%) | 12/41 patients (29.3%) | 0.85 |
| **Troponin T mean** | All patients | 97.4 ng/L | 9/14 patients (64.3%) | 31/81 patients (38.3%) | 1.68 |
| | Only patients without ECMO | | 4/8 patients (50.0%) | 9/38 patients (23.7%) | 2.11 |

**Table 2. Relative mortality in the second-wave cohort (multivariable analysis).** Using the cut-off value for a survival probability of 50%, which had been determined by multivariabe regression analysis for the first-wave patients, the cohort was divided into two groups: patients with values above and patients with values below the cut-off. Mortality within the groups and, from these results, the relative mortality was calculated. pHmin, minimum pH of blood during the 14-day observation period for each patient; MAPmean, mean MAP during the 14-day observation period for each patient.

| | | Cut-off | Absolute mortality of patients with values below (MAP, pH, BE) or above (Troponin) cut-off | Absolute mortality of patients with values above (MAP, pH, BE) or below (Troponin) cut-off | Relative mortality |
|---|---|---|---|---|---|
| **pH min+ 8.37*10$^{-3}$*mmHg$^{-1}$*MAPmean** | All patients | 7.93 | 12/17 patients (70.6%) | 29/81 patients (35.8%) | 1.97 |
| | Only patients without ECMO | | 6/9 patients (66.7%) | 8/40 patients (20.0%) | 3.33 |

50%. To apply this model to the patients of the second-wave cohort, a combined parameter pHmin+8.37*10$^{-3}$*mmHg$^{-1}$*MAPmean was calculated for each patient, and the mortality rate was determined from this calculation (Table 2). For first-wave patients a relative mortality of 6.92 had been calculated this way.

Sensitivity, specificity, as well as positive and negative prognostic values were calculated for all models and are summarized in S4 Appendix.

In the multivariable model, the same pattern was seen as for the univariable models of MAP, pH, and troponin: when only considering patients without ECMO therapy, outcome was correctly estimated in 77.6% of the patients, but this proportion decreased when including the entire cohort.

This pattern also becomes clear when looking at the values for pHmin and MAPmean for each patient in a plane spanned by these parameters (Fig 2). The calculated cut-off value for the combination of these two values results in a straight line in the plane; this line divides the patients of the first-wave cohort (Fig 2A) and the patients without ECMO therapy of the second-wave cohort (Fig 2C) into two groups, in which the probability of death calculated by the multivariable model was greater or less than 0.5: As clearly visible, only a few patients were not correctly assigned with respect to their actual outcome. In contrast, for patients of the second-wave cohort who received ECMO therapy, the assignment did not work nearly as well because this patient group apparently shifted towards lower MAP values than patients without ECMO therapy.

## Discussion

Mortality rates did not significantly differ between the second-wave cohort in this study and the first-wave cohort of the previous study. This finding is consistent with the data provided by German register rates [8] and with data from the neighboring country France [9], even though some centers in Germany apparently had lower mortality rates during the second wave [10].

The comparison of the second-wave cohort with the first-wave cohort yielded the following results: In the first-wave cohort, pH of blood, MAP, BE, troponin, and PCT had been identified as highly significant prognostic factors of in-hospital mortality. In the second-wave cohort, survivors and non-survivors differed highly significantly in pH of blood and troponin and significantly in MAP and PCT but not in BE. On the other hand, survivors and non-survivors in the second-wave cohort showed significant or highly significant differences in pulmonary gas exchange and ventilation parameters in contrast to patients of the first-wave cohort. In particular, survivors and non-survivors in the second-wave cohort differed highly significantly in $SpO_2$, $paCO_2$, $FiO_2$ and the P/F ratio.

The question now arises as to the reason for the differences between these two cohorts. In principle, three explanatory approaches would be plausible: First, the patient characteristics in

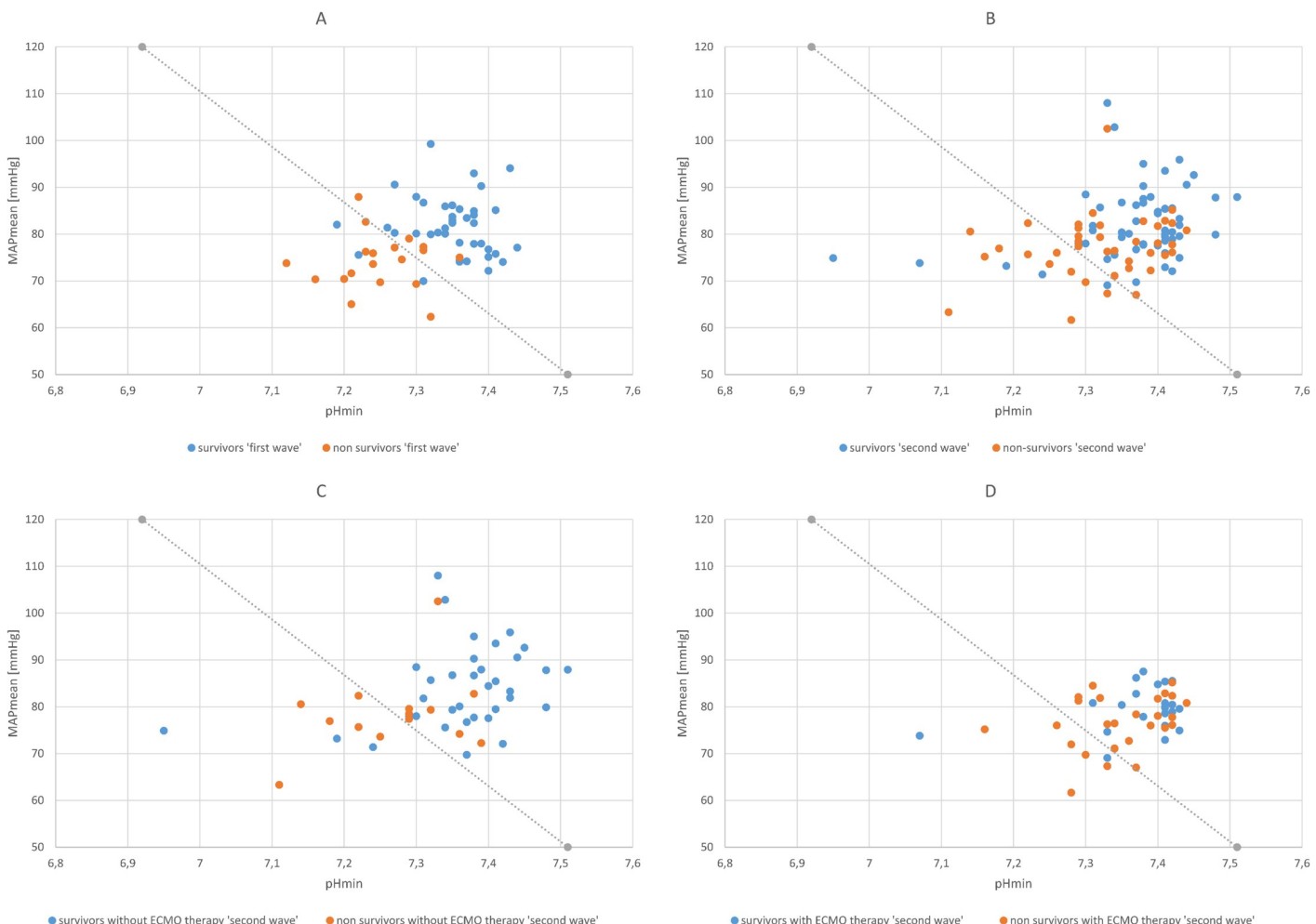

**Fig 2. Visualization of the prognostic power of the multivariable regression model and comparison of the first-wave cohort with the second-wave cohort.** The cut-off line drawn in the diagrams separates the plane spanned by pHmin and MAPmean into a region below this line with a survival probability <0.50 and into a region above this line with a survival probability >0.50. The values calculated for each patient are plotted in the diagram. 2A: All patients of the first-wave cohort. The cut-off line separates survivors and non-survivors with high accuracy. 2B: All patients of the second-wave cohort. The majority of non-survivors appear to be above the cut-off line, which means that many non-survivors would have been falsely prognosed to survive. 2C: Patients with ECMO therapy of the second-wave cohort. In this subgroup, survivors and non-survivors are again separated by the cut-off line with high accuracy. 2D: Patients with ECMO therapy of the second-wave cohort. A useful separation of survivors and non-survivors by the cut-off line is not possible. In the group of non-survivors, there seems to be a shift towards higher values for pHmin. pHmin, minimum pH of blood during the 14-day observation period for each patient; MAPmean, mean MAP during the 14-day observation period for each patient.

the two cohorts were fundamentally different. Second, the underlying mutation variants of the virus differed between the first-wave and the second-wave cohort, resulting in different courses of disease. Third, the course of disease at the ICU during the second wave was influenced by different or new treatment strategies.

The comparison of the two cohorts shows that baseline characteristics did not differ between the first-wave and the second-wave cohort, but patients of the second-wave cohort had on average more pre-existing co-morbidities than patients of the first-wave cohort. In addition, the duration of pre-treatment at an external ICU before the transfer to the University Medical Center Regensburg was also significantly longer in the second-wave cohort. Thus, it is conceivable that, due to the longer pre-treatment duration at an external ICU, the investigated first 14 days of intensive care treatment at our hospital actually corresponded to a later stage of disease in the patients of the second-wave cohort than in the patients of the first-wave cohort.

By now, several studies have reported on the fact that different mutational variants of the virus causing COVID-19 [11, 12] most recently for the variant B.1.1.529 ('omicron') [13], may be associated with different disease severity. In September 2020, the virus mutation variant B.1.1.7 ('alpha') was identified for the first time in the UK [14]. This variant showed significant differences to previous mutation variants in terms of transmissibility [11] but also in disease progression and case fatality [15, 16]. Yet in Germany, the rate of proven infections with the alpha mutation variant was only 2.2% in the first calendar week of 2021 and remained below 20% until mid-February 2021 but predominated shortly thereafter during the third wave in Germany [17]. The viral mutation variants B.1.351 ('beta'), B.1.617.2 ('delta'), and P.1 ('gamma'), which—like alpha—are considered Variants of Concern (VOCs) according to the World Health Organization (WHO) [18], were hardly detected at all during the study period [19]. Even though sequencing of the viral mutation variant was only available for a very small proportion of the patients included in our evaluation, it can be assumed that only a minority of patients were infected with the alpha variant. It could be shown that the outbreak of COVID-19 in Germany at the turn of 2020–2021 was not dominated by a single variant. Furthermore, the largest proportion of this outbreak was caused by the variant B.1.177, which is neither considered to be conspicuous in terms of its characteristics [20] nor is one of the VOCs. Accordingly, the first-wave and second-wave cohorts should be comparable in terms of basic disease course and basic disease severity.

Since the publication of the RECOVERY trial [6], critical care patients with COVID-19 have been treated with dexamethasone regularly. Although the results of that study were not published until February 2021, critical care patients in Germany seemed to have already received corticosteroids more frequently during the second wave than during the first wave [10], which may have influenced the course of ICU treatment.

Of note is the increased use of ECMO therapy in the second-wave cohort. In the initial phase of the pandemic, ECMO therapy had apparently been used rather cautiously because of the high mortality rates published for patients receiving ECMO therapy [21] and the restrictive initial recommendations of the Extracorporeal Life Support Organization (ELSO) [22]. In contrast, about 50% of the patients in our second-wave cohort received ECMO therapy. This result is in line with published data for Europe [23], although the prognosis of patients requiring ECMO therapy shows progressively worse survival over time [24]. In our current study, significantly more non-survivors than survivors received ECMO therapy. A major reason for using veno-venous ECMO therapy is to reduce the invasiveness of controlled ventilation. Therefore, it seems at least plausible that such a reduction may lead to differences in ventilation parameters. The same assumption can be made for pulmonary gas exchange because the use of veno-venous ECMO therapy should improve both, oxygenation and decarboxylation. The supposition that the more frequent use of ECMO therapy in the second-wave cohort may have been a major reason for the different results of some parameters between the two cohorts fits with our findings. Our current study shows that the models for prognosing lethal outcome calculated with data from the first-wave cohort can also be used with high accuracy for the second-wave cohort, but only for patients who did not receive ECMO therapy.

## Limitations

A general limitation of this study is its low number of patients; thus, larger-scale studies are needed to confirm the presented results.

As already stated in the limitations section of the previous work, the patients included in the present study were treated at a university hospital, hence at the highest level of ICU therapy

available. Therefore, the findings may possibly be not transferrable one-by-one to situations in smaller hospitals with more limited resources and logistic options.

Our study did not investigate specific effects of the use of ECMO therapy in patients with COVID-19, so only limited information is available on this topic. We have already started a follow-up study on the course of ICU treatment in this specialized patient population.

When only considering patients without ECMO therapy, the case number of the second-wave cohort is rather low (n = 49), which means that the power of statistical calculations is limited.

## Conclusions

We could show that the models for prognosing a lethal course in critical care patients with COVID-19 that had been calculated for patients of the first wave by means of univariable and multivariable regression analysis also have good prognostic power in patients of the second wave, but only for patients who did not receive ECMO therapy.

## Supporting information

**S1 Appendix. Ethics approval document.**
(PDF)

**S2 Appendix. Description of the derivation of the combined cut-off value for the multivariable model.**
(DOCX)

**S3 Appendix. Minimal data set: Tabular overview of the data collected for each patient.**
(XLSX)

**S4 Appendix. Calculation of sensitivity, specificity, as well as positive and negative prognostic values for all parameters and cut-offs presented in Tables 1 and 2.**
(DOCX)

**S1 Table. List of all observed parameters.**
(DOCX)

**S2 Table. Overview of all analyzed parameters recorded on a daily basis for patients during the second wave and the first wave.**
(DOCX)

**S1 Fig. Visualization of vital signs.**
(PDF)

**S2 Fig. Dosage of catecholamines, analgosedation, anticoagulation, and antithrombotic medication.**
(PDF)

**S3 Fig. Visualization of laboratory blood diagnostics and microbiological diagnostics.**
(PDF)

**S4 Fig. Ventilator therapy and pulmonary gas exchange.**
(PDF)

**S5 Fig. Extracorporeal membrane oxygenation (ECMO) and renal replacement therapy (RRT).**
(PDF)

## Author Contributions

**Conceptualization:** Martin Kieninger, Matthias Lubnow, Bärbel Kieninger.

**Formal analysis:** Martin Kieninger, Sarah Dietl, Annemarie Sinning, Wolfram Gronwald, Florian Zeman, Bärbel Kieninger.

**Investigation:** Martin Kieninger, Sarah Dietl, Annemarie Sinning, Bärbel Kieninger.

**Methodology:** Martin Kieninger, Wolfram Gronwald, Florian Zeman, Bärbel Kieninger.

**Software:** Florian Zeman, Bärbel Kieninger.

**Supervision:** Martin Kieninger.

**Validation:** Michael Gruber, Wolfram Gronwald, Florian Zeman, Bernd Salzberger.

**Visualization:** Martin Kieninger, Sarah Dietl, Bärbel Kieninger.

**Writing – original draft:** Martin Kieninger, Bärbel Kieninger.

**Writing – review & editing:** Michael Gruber, Dirk Lunz, Thomas Dienemann, Stephan Schmid, Bernhard Graf, Matthias Lubnow, Thomas Müller, Thomas Holzmann, Bernd Salzberger.

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
