## [Decision Letter · Decision Letter 0]

28 Feb 2022

PONE-D-21-39783Evaluation of models for prognosing mortality in critical care patients with COVID-19: First- and second-wave data from a German university hospitalPLOS ONE

Dear Dr. Kieninger,

Thank you for submitting your manuscript to PLOS ONE. After careful consideration, we feel that it has merit but does not fully meet PLOS ONE’s publication criteria as it currently stands. Therefore, we invite you to submit a revised version of the manuscript that addresses the points raised during the review process.

We look forward to receiving your revised manuscript.

Kind regards,

Yu-Chang Yeh, M.D., Ph.D.

Academic Editor

PLOS ONE

Journal Requirements:

Reviewers' comments:

Reviewer's Responses to Questions

**Comments to the Author**

1. Is the manuscript technically sound, and do the data support the conclusions?

Reviewer #1: Yes

Reviewer #2: Partly

2. Has the statistical analysis been performed appropriately and rigorously? 

Reviewer #1: Yes

Reviewer #2: No

3. Have the authors made all data underlying the findings in their manuscript fully available?

Reviewer #1: Yes

Reviewer #2: Yes

4. Is the manuscript presented in an intelligible fashion and written in standard English?

Reviewer #1: Yes

Reviewer #2: No

5. Review Comments to the Author

Reviewer #1: The authors describe a retrospective cohort study on 98 critical care patients with COVID-19 that analyzed if a previously developed prognostic model from a previous wave was valid in a subsequent wave.

The statement in the introduction on page 3, reads, "These waves were triggered by both various lock-down measures and the spread of different virus variants". I do not understand how waves could be triggered by lock-down measures. I would suggest that lock down public health measures were triggered by the waves.

The methodology was appropriate and clearly described. Statistical analysis was appropriate. The findings are well described and the figures and tables are appropriate and understandable. The conclusions are supported by the data.

The limitations should add that the number of patients is small and that larger studies should be performed.

The references are relevant and current.

Reviewer #2: Evaluation of models for prognosing mortality in critical care patients with COVID-19: First- and second-wave data from a German university hospital.

The present study by Kieninger et al. aims to evaluate a prognostic model for outcome assessment of ICU patients admitted for COVID-19. Data were examined during the first 2 weeks of ICU treatments and patients were followed until death or ICU discharge.

Even if the topic is interesting, some points should be clarified:

- Line 128. It is not clear in the text the presence of the first and second wave data in the table. I encourage the authors to clarify this point in the text.

- Line 130. Why cut-off values were not based on ROC curves with values derives from best sensitivity and specificity? On the contrary, why you used as cut-off values the numbers associated with 50% probability of the event, as reported in the S2 appendix. Accuracy cannot be measured in this model. A probability cut-off of 0.5 is usually not appropriate since the output could be heavily skewed in real word data. You should clarified these points.

- Supplementary figures: please name the axes of the graphs

- Line 386 the possible role of virus variants for disease time course should be adequately discussed and referred. Moreover, as stated at line 406-408 no major differences were observed in term of virus mutation in the 2 cohorts.

- The authors should clarify the reason of using the first 14 days of ICU data for the model development. The mortality rate is steeper in the first 30 days and a prediction model should predict the outcome much earlier than shortly before the event happened.

- A direct comparison for relative mortality rates between the two waves should be provided.

- Almost sensitivity, specificity along with PPV, NPV and ROC curve of the model should be provided

6. PLOS authors have the option to publish the peer review history of their article (what does this mean?). If published, this will include your full peer review and any attached files.

Reviewer #1: No

Reviewer #2: **Yes: **Matteo Leoni

---

## [Author Response · Author response to Decision Letter 0]

18 Mar 2022

PONE-D-21-39783R1

Revised version of the manuscript Evaluation of models for prognosing mortality in critical care patients with COVID-19: First- and second-wave data from a German university hospital

Dear Yu-Chang Yeh,

On behalf of my co-authors, I would like to thank you and the reviewers for your valuable comments and suggestions to improve the quality of our manuscript. We have revised the manuscript according to the reviewers' comments. Enclosed you find a point-by-point response to each comment indicating the action taken or the revision made.

In addition to this letter, we have uploaded a version of the revised manuscript with tracked changes ('Revised Manuscript with Track Changes') and a version without tracked changes ('Manuscript').

Besides addressing the reviewers’ comments, we have also dealt with the following issues:

1. PLOS ONE's style requirements, including those for file naming,

2. Data availability: We have added a supporting information file 'S3 Appendix', which contains the minimal data set underlying the results described in our manuscript, and

3. All figure files uploaded to the Editorial Manager have been checked and adjusted with the Preflight Analysis and Conversion Engine (PACE) digital diagnostic tool.

Reviewer #2 noted that the manuscript was not presented in an intelligible fashion and was not written in standard English. We would like to point out that the manuscript has been revised by a professional language editor (Mrs. Monika Schöll, B.A., Open University, Milton Keynes, UK).

When reviewing the manuscript again, we noticed that some values in Table 1 were slightly incorrect. These values have been corrected. We would like to emphasize that this oversight did not affect any of the results or statements made in the paper in any way.

Thank you very much again for your help in improving the quality of our paper. We trust that our manuscript is now suitable for publication in PLOS ONE.

Sincerely,

PD Dr. Martin Kieninger

 

Reviewer #1:

Comment 1: 

The statement in the introduction on page 3, reads, "These waves were triggered by both various lock-down measures and the spread of different virus variants". I do not understand how waves could be triggered by lock-down measures. I would suggest that lock down public health measures were triggered by the waves.

Response: 

We have rephrased the sentence (line 56-57):

These waves were primarily triggered by the spread of different virus variants.

Comment 2: 

The limitations should add that the number of patients is small and that larger studies should be performed.

Response: 

We have added the following sentence to the limitations section (line 452-453):

A general limitation of this study is its low number of patients; thus, larger-scale studies are needed to confirm the presented results.

Reviewer #2:

Comment 1: 

Line 128. It is not clear in the text the presence of the first and second wave data in the table. I encourage the authors to clarify this point in the text.

Response: 

In order to clarify the facts, we have further elaborated the explanations in two places in the text (line 92-95 and line 132-135).

This retrospective study included newly acquired data from 98 critically ill adult patients with COVID-19 (76 men, 22 women), who had been treated at one of the ICUs at the University Medical Center Regensburg between October 2020 and February 2021 (second-wave) and comparative data from 59 patients of the first wave, as already published in (5).

In S2 Table, data from the first wave were also included, and differences between survivors and non-survivors were assessed by categorization across the defined significance levels.

Comment 2: 

Line 130. Why cut-off values were not based on ROC curves with values derives from best sensitivity and specificity? On the contrary, why you used as cut-off values the numbers associated with 50% probability of the event, as reported in the S2 appendix. Accuracy cannot be measured in this model. A probability cut-off of 0.5 is usually not appropriate since the output could be heavily skewed in real word data. You should clarified these points.

Response: 

The aim of this follow-up study was to evaluate the results of the first study in a second cohort. Therefore, we re-used the cut-off values already established in the first study to see to what extent these values may distinguish survivors from non-survivors in the second wave.

The primary goal of the first study was to identify parameters related to ICU therapy that could contribute to predicting outcome. The models were the mathematical way to achieve this aim. We may refer to the material and methods section of the first study.

The question why we did not derive the cut-off values in the first study by optimizing sensitivity and specificity is of course justified. Given the low number of patients available, however, optimizing cut-off values did not seem mathematically reasonable. In addition, calculating sensitivity and specificity for the parameters with the chosen cut-off values showed a tendency towards better specificity in each case (to the disadvantage of sensitivity). This aspect was important to us because, when applying the results to an extreme situation in which a final decision has to be made about continuing or discontinuing ICU treatment, false prediction of non-survival has to be avoided.

Comment 3: 

Supplementary figures: please name the axes of the graphs

Response:

We have named the axes of all graphs in the Supporting information files (S1 Figure, S2 Figure, S3 Figure, S4 Figure, and S5 Figure).

Comment 4: 

Line 386 the possible role of virus variants for disease time course should be adequately discussed and referred. Moreover, as stated at line 406-408 no major differences were observed in term of virus mutation in the 2 cohorts.

Response: 

We have revised the section on the possible role of viral variants in disease progression in the discussion section (line 410-426):

By now, several studies have reported on the fact that different mutational variants of the virus causing COVID-19 (11, 12) most recently for the variant B.1.1.529 (‘omicron’) (13), may be associated with different disease severity. In September 2020, the virus mutation variant B.1.1.7 ('alpha') was identified for the first time in the UK (14). This variant showed significant differences to previous mutation variants in terms of transmissibility (11) but also in disease progression and case fatality (15, 16). Yet in Germany, the rate of proven infections with the alpha mutation variant was only 2.2% in the first calendar week of 2021 and remained below 20% until mid-February 2021 but predominated shortly thereafter during the third wave in Germany (17). The viral mutation variants B.1.351 ('beta'), B.1.617.2 ('delta'), and P.1 ('gamma'), which—like alpha—are considered Variants of Concern (VOCs) according to the World Health Organization (WHO) (18), were hardly detected at all during the study period (19). Even though sequencing of the viral mutation variant was only available for a very small proportion of the patients included in our evaluation, it can be assumed that only a minority of patients were infected with the alpha variant. It could be shown that the outbreak of COVID-19 in Germany at the turn of 2020-2021 was not dominated by a single variant. Furthermore, the largest proportion of this outbreak was caused by the variant B.1.177, which is neither considered to be conspicuous in terms of its characteristics (20) nor is one of the VOCs. Accordingly, the first-wave and second-wave cohorts should be comparable in terms of basic disease course and basic disease severity.

In addition, we would like to point out that some of the authors themselves worked on the sequencing of SARS-CoV-2, using samples of patients in the area of Eastern Bavaria and thus from the catchment area of our hospital. By the end of 2021, the variant 'alpha' could not be found in these probes.

Comment 5: 

The authors should clarify the reason of using the first 14 days of ICU data for the model development. The mortality rate is steeper in the first 30 days and a prediction model should predict the outcome much earlier than shortly before the event happened.

Response: 

Again, we chose the same method of data collection for this study as we did for our first study in order to have a congruent situation and to be able to check the value of the originally calculated models in patients of the second wave.

Of course, it would be desirable to have valid predictors for the further course of a patient's treatment already at the time of admission to the intensive care units. This is difficult, if not impossible, not least because of the very different pretreatment of the patients. Many of the patients included in this study had been transferred to our university hospital from non-tertiary hospitals with often limited treatment options. From our point of view, the chosen observation period of 2 weeks seems therefore reasonable in this context. If a patient's condition has not improved after two weeks of maximum intensive medical therapy, the question usually arises as to whether it makes sense to continue intensive medical therapy. Also, in view of the fact that critically ill patients with COVID-19 often have to be treated in an intensive care unit for many weeks, a re-evaluation of the situation after two weeks makes sense according to our clinical experience. Therefore, we assume that the calculated model can make a valuable contribution to decision-making in these complex, critically ill patients.

Comment 6: 

A direct comparison for relative mortality rates between the two waves should be provided.

Response: 

We have made the following additions to the manuscript (line 312-314 and line 338):

In first-wave patients included in the previous study, relative mortality was 2.64 for MAPmean, 7.89 for pHmean, 3.57 for pHmax, 5.03 for pHmin, 2.57 for BEmean, 3.25 for BEmax, and 2.88 for troponin Tmean.

For first-wave patients a relative mortality of 6.92 had been calculated this way.

Comment 7: 

Almost sensitivity, specificity along with PPV, NPV and ROC curve of the model should be provided.

Response: 

As suggested by the reviewers of the first study, we had presented the relevance of the parameters for distinguishing survivors from non-survivors based on the relative mortality of the groups to each other and have now repeated this procedure for comparability of results (Table 1, Table 2).

As suggested by reviewer #2 we have now additionally calculated sensitivity, specificity, as well as positive and negative prognostic values for all parameters and cut-offs presented in Table 1 and Table 2 of the present study, respectively (S4 Appendix).

We have added the following sentence to the manuscript (line 348-349):

Sensitivity, specificity, as well as positive and negative prognostic values were calculated for all models and are summarized in S4 Appendix.

---

## [Decision Letter · Decision Letter 1]

18 Apr 2022

PONE-D-21-39783R1Evaluation of models for prognosing mortality in critical care patients with COVID-19: First- and second-wave data from a German university hospitalPLOS ONE

Dear Dr. Kieninger,

Thank you for submitting your manuscript to PLOS ONE. After careful consideration, we feel that it has merit but does not fully meet PLOS ONE’s publication criteria as it currently stands. Therefore, we invite you to submit a revised version of the manuscript that addresses the points raised during the review process.

We look forward to receiving your revised manuscript.

Kind regards,

Yu-Chang Yeh, M.D., Ph.D.

Academic Editor

PLOS ONE

Journal Requirements:

Reviewers' comments:

Reviewer's Responses to Questions

**Comments to the Author**

1. If the authors have adequately addressed your comments raised in a previous round of review and you feel that this manuscript is now acceptable for publication, you may indicate that here to bypass the “Comments to the Author” section, enter your conflict of interest statement in the “Confidential to Editor” section, and submit your "Accept" recommendation.

Reviewer #1: (No Response)

Reviewer #2: All comments have been addressed

2. Is the manuscript technically sound, and do the data support the conclusions?

Reviewer #1: (No Response)

Reviewer #2: Yes

3. Has the statistical analysis been performed appropriately and rigorously? 

Reviewer #1: Yes

Reviewer #2: Yes

4. Have the authors made all data underlying the findings in their manuscript fully available?

Reviewer #1: Yes

Reviewer #2: Yes

5. Is the manuscript presented in an intelligible fashion and written in standard English?

Reviewer #1: Yes

Reviewer #2: Yes

6. Review Comments to the Author

Reviewer #1: The authors have addressed my comments and that of the other reviewers. My only new comment is that the authors should use the term ventilator or mechanical ventilator rather than respirator, which, typically, refers to a mask such as an N95 or FFP2 respirator.

Reviewer #2: I appreciated the answers provided by the authors. Outcome prediction is a very interesting topic and I think the article is ready for publication.

7. PLOS authors have the option to publish the peer review history of their article (what does this mean?). If published, this will include your full peer review and any attached files.

Reviewer #1: No

Reviewer #2: **Yes: **Matteo L.G. Leoni

---

## [Author Response · Author response to Decision Letter 1]

2 May 2022

Reviewer #1:

Comment 1: 

The authors have addressed my comments and that of the other reviewers. My only new comment is that the authors should use the term ventilator or mechanical ventilator rather than respirator, which, typically, refers to a mask such as an N95 or FFP2 respirator.

Response: 

We have replaced the term 'respirator' with 'ventilator' in all affected places in the manuscript.

---

## [Editor Report · Decision Letter 2]

9 May 2022

Evaluation of models for prognosing mortality in critical care patients with COVID-19: First- and second-wave data from a German university hospital

PONE-D-21-39783R2

Dear Dr. Kieninger,

We’re pleased to inform you that your manuscript has been judged scientifically suitable for publication and will be formally accepted for publication once it meets all outstanding technical requirements.

Kind regards,

Yu-Chang Yeh, M.D., Ph.D.

Academic Editor

PLOS ONE
---

## [Editor Report · Acceptance letter]

17 May 2022

PONE-D-21-39783R2 

Evaluation of models for prognosing mortality in critical care patients with COVID-19: First- and second-wave data from a German university hospital 

Dear Dr. Kieninger:

I'm pleased to inform you that your manuscript has been deemed suitable for publication in PLOS ONE. Congratulations! Your manuscript is now with our production department. 

Kind regards, 

on behalf of

Dr. Yu-Chang Yeh 

Academic Editor

PLOS ONE